# Effects of Carbonated Thickened Drinks on Pharyngeal Swallowing with a Flexible Endoscopic Evaluation of Swallowing in Older Patients with Oropharyngeal Dysphagia

**DOI:** 10.3390/healthcare10091769

**Published:** 2022-09-14

**Authors:** Motoyoshi Morishita, Masahiko Okubo, Tatsuro Sekine

**Affiliations:** 1Department of Physical Therapy, Kibi International University, 8, Iga-machi, Takahashi 716-8508, Okayama, Japan; 2Department of Oral and Maxillofacial Surgery, Faculty of Medicine, Saitama Medical University, 38, Morohongo, Moroyama-machi 350-0475, Saitama, Japan; 3Department of Dentistry, Yokohama Izumidai Hospital, 7838, Izumi-cho, Izumi-ku, Yokohama 245-0016, Kanagawa, Japan; 4Department of Otolaryngology, Faculty of Medicine, Saitama Medical University, 38, Morohongo, Moroyama-machi 350-0475, Saitama, Japan

**Keywords:** dysphagia, carbonated thickened drink, flexible endoscopic evaluation of swallowing, subjective difficulty of swallowing

## Abstract

This study aimed to determine the efficacy of carbonated and sweetened drinks added to thickened liquids, which are routinely used for patients with dysphagia to improve dysphagia. Patients swallowed thin liquid (Thin), thickened liquid (Thick), carbonated thin drink (C-Thin), and carbonated thickened drink (C-Thick) in random order. Penetration and/or aspiration were scored using the Penetration–Aspiration Scale (PAS). The residue was scored using the Yale Pharyngeal Residue Severity Rating Scale (YPR-SRS). Swallowing reflex initiation was scored using the Hyodo score. The subjective difficulty of swallowing was scored on a face scale. We analyzed 13 patients with a mean age of 79.6 ± 9.6 years. PAS was significantly lower in the C-Thick group than the Thin group (*p* < 0.05). Swallowing reflex initiation was significantly different between the Thin and Thick (*p* < 0.01) groups; moreover, post hoc analysis revealed that it was significantly lower in the C-Thick group than the Thin group (*p* < 0.01). The subjective difficulty of swallowing in the C-Thick was significantly lower than the Thick group (*p* < 0.05). C-Thick was easier to swallow than Thick and may improve penetration and/or aspiration in older patients with dysphagia with complex diseases.

## 1. Introduction

Sensory stimulation of the oral cavity and pharynx are widely used to aid the swallowing activity of patients with dysphagia. As physical stimuli, cold and tactile stimulation of the oral cavity and pharynx are often used in patients with dysphagia who have problems in the pharyngeal and laryngeal phases of swallowing. These stimuli stimulate the swallowing reflex by lowering the threshold of receptors in the anterior palatine arch and increasing afferent nerve input to the medulla oblongata [1]. Taste stimuli to taste receptor cells on the tongue also influence swallowing activity. Taste stimuli, especially sour stimuli, increase swallowing pressure and muscle activity of the suprahyoid muscles [2,3]. In addition, salty and sweet stimuli increase swallowing pressure and suprahyoid muscle activity compared to no taste stimuli [2,3]. Sweet taste stimulation is also an important facilitator of swallowing, as it excites and inhibits the nucleus tractus solitarius and modulates the swallowing motor pathways in the cerebral cortex [4].

Carbonated water is also a sensory stimulus of the oral cavity and pharynx. Carbonated water stimulates nociceptors in the oral mucosa and activates the trigeminal nerve when carbon dioxide (CO_2_) dissolved in water reacts with carbonic anhydrase IV (CA-IV) in salivary enzymes to produce carbonic acid (H_2_CO_3_) [5,6,7]. Furthermore, taste receptors that detect stimulation by carbonated water stimulate acidity-sensitive taste receptor cells via the facial nerve when the carbonic acid in carbonated water is divided into bicarbonate ions and free protons in the oral cavity [7]. Other studies have shown that CO_2_ may act on the taste system and other sensory pathways, such as by strongly stimulating the somatosensory system, indicating that the carbonic acid perception system is complex [7].

A recent meta-analysis showed that carbonated water reduced aspiration and increased swallowing apnea duration compared to non-carbonated liquids [8]. However, another study reported that the evidence remains inconclusive given the limited overlap in methodologies and reduced ability to compare findings within and between studies [9].

Although most previous studies have been conducted with carbonated water, the previous study suggested that sweetened carbonated beverages were easier to swallow than unsweetened carbonated water for Japanese subjects [10]. Furthermore, more palatable foods strengthen the swallowing-related networks in the cerebrum [11]. Therefore, carbonated beverages may be more effective than carbonated water in improving swallowing.

Thickened liquids are widely used in clinical practice to treat patients with dysphagia [12]. Thin liquids may enter the larynx before the airway closes during swallowing in patients with dysphagia. Thickness is often recommended with the aim of slowing the flow of liquid to bring more time for airway closure [13]. Thus, carbonated thickened liquid may be more effective than carbonated thin liquid. The use of carbonated thickened liquid may be feasible as a new technique for safe fluid intake in patients with dysphagia, for whom thickened liquids or carbonated thin liquids have caused aspiration in the past. However, no studies on the swallowing behavior of carbonated thickened liquids have been reported previously.

Our study aimed to determine the effect of sweetened carbonated thickened drink on penetration and/or aspiration, pharyngeal residue, and swallowing reflex in older patients with dysphagia with complex diseases, using the flexible endoscopic evaluation of swallowing (FEES) test and to identify differences in swallowing behavior versus other liquids. 

## 2. Materials and Methods

This study was a cross-sectional study. Patients who underwent FEES for the diagnosis of dysphagia at Yokohama Izumidai Hospital between 2021 and 2022 were included in this study. Exclusion criteria were age less than 18 years and subjects with cognitive dysfunction that prevented hearing and subjective difficulty swallowing liquids. The Ethical Review Committee of the corresponding author’s institution approved the study (Approval number: 21-03). We registered our experiment’s protocol in the University hospital Medical Information Network (UMIN) clinical trial registration system (The trial registration number: UMIN000048144). Data including sex, age, underlying disease, etiology of dysphagia, and functional oral intake score (FOIS) [14] were collected for all subjects.

Patients swallowed the amount they could swallow in one mouthful (5–15 cc) from a cup filled with one of four liquids cooled to approximately 10°C in random order. The liquids were thin liquid (Thin) (water), thickened liquid (Thick) (water), carbonated thin drink (C-Thin) (sweet taste), and carbonated thickened drink (C-Thick) (sweet taste). Each swallow was followed by a 5-minute interval.

For C-Thick, we adjusted the amounts of xanthan gum 0.05–0.50% (*w*/*w*) and tamarind seed gum 0.20–1.00% (*w*/*w*) to achieve a viscosity of 100 mPa∙s (nectar-thick). We dissolved 0.03% (*w*/*w*) acesulfame potassium and 0.01% (*w*/*w*) sucralose for sweetening and 0.10% (*w*/*w*) citric acid and 0.02% (*w*/*w*) trisodium citrate for souring. Additionally, we dissolved 0.05% (*w*/*w*) Brilliant Blue FCF (FD & C Blue No. 1) and 0.05% (*w*/*w*) orange-lime-lemon flavoring. These materials were dissolved using a hand mixer. Viscosity was measured using a cone-plate rotational viscometer (Viscometer TV-25, Toki Sangyo, Japan) at a temperature of 20 °C and a shear rate of 50 s^−1^ after 1 min. The liquid was placed in a tank that was pressurized with carbon dioxide. The pressure was adjusted to reach a gas volume of 2.0, and carbonation was added. The liquid was filled into containers, sealed, and heat pasteurized. After pasteurization, the gas volume was measured using a gas volume measuring device (Air tester 5001, Zahm & Nagel, Holland, NY, USA). For C-Thin, we dissolved 0.03% (*w*/*w*) acesulfame potassium and 0.01% (*w*/*w*) sucralose for sweetening and 0.10% (*w*/*w*) citric acid and 0.02% (*w*/*w*) trisodium citrate for souring. Additionally, we dissolved 0.05% (*w*/*w*) Brilliant Blue FCF (FD & C Blue No. 1) and 0.05% (*w*/*w*) orange-lime-lemon flavoring. The carbonic acid dissolution procedure and gas volume measurements were performed using the same process as for C-Thick. For Thick, we adjusted tap water in the amounts of xanthan gum 0.05–0.50% (*w*/*w*) and tamarind seed gum 0.20–1.00% (*w*/*w*) to achieve a viscosity of 100 mPa∙s (nectar-thick). Viscosity was measured using the same procedure as for C-Thick.

The patient was inserted through the nasal cavity with a flexible digital video-rinolaryngoscope (Pentax portable multi scope FP-7RBS2, Pentax, Tokyo, Japan) in a sitting position. No anesthesia was used during nasal insertion. Gel (Okuchi o arau gel, Nippon Shika Yakuhin Co., Ltd., Shimonoseki, Japan) was coated on the laryngoscope. A 2-hour pre-test dietary restriction was imposed before FEES. No sputum was suctioned prior to FEES.

Pharyngeal residue, aspiration, and swallowing reflex initiation were evaluated based on FEES findings at the time of swallowing each liquid. Secretion status was scored before swallowing using the Murray secretion scale (MSS), which is scored on a scale of 0 to 3, with 0 meaning “No visible secretions anywhere in the hypopharynx or some transient bubbles visible in the vallecula and pyriform sinuses”. A higher MSS score indicates a more severe secretion status [15]. Pharyngeal residue was scored by the Yale pharyngeal residue severity rating scale (YPR-SRS), which was used to assess vallecula residue and pyriform sinus residue. In the YPR-SRS, 1 means “No residue”, and 5 means “Filled to epiglottic rim” or “Filled to aryepiglottic fold” [16].

Aspiration was assessed with the Penetration–Aspiration Scale (PAS). Aspiration is scored as 6, 7, or 8. Penetration, on the other hand, is scored as either 2 or 3 if residue remains above the vocal folds, and 4 or 5 if residue courses to the level of the vocal folds. Normal is scored as 1. The swallowing reflex initiation was assessed from FEES using the Hyodo scoring method. The location of the bolus at the swallowing reflex initiation was assessed by “white-out” timing [17]. The site of the swallowing reflex initiation is scored as 0 for pharyngeal, 1 for Valleculla, 2 for Piriform sinus, and 3 when the swallowing reflex is not elicited.

FEES was recorded and scored by a dentist with more than 10 years of FEES experience. The patient’s subjective difficulty in swallowing other liquids compared to Thin was recorded using a face scale (Figure 1). In patients with a FOIS score of 1–2, the following conditions were confirmed before measurement to minimize the risk for aspiration. (1) Clear level of consciousness; (2) cognitive function to report difficulty swallowing and distress in swallowing; (3) can swallow saliva voluntarily as instructed; (4) has no obvious xerostomia; and (5) has an MSS score of 0 or 1 before swallowing liquid.

Statistical analysis was performed using SPSS Ver. 23 (SPSS, IBM, Co., Ltd., Armonk, NY, USA). Friedman test was used for comparisons for PAS, YPR-SRS, subjective difficulty of swallowing, and the swallowing reflex initiation per liquid, and the Bonferroni method was used for multiple comparisons. In multiple comparisons, the effect size (r) was determined from the statistic index value (Z value) for all significant pairs. The resulting absolute r value was interrupted, with 0.3–0.5 indicating a medium effect and ≥0.5 indicating a large effect. The relationship among PAS, YPR-SRS, subjective difficulty of swallowing, and the swallowing reflex initiation per liquid was examined using Spearman’s rank correlation coefficient.

The patient indicated difficulty swallowing compared to Thin by pointing.

## 3. Results

### 3.1. Population

Of the thirteen patients, eight were male, and the mean age of the patients was 79.6 ± 9.6 years (Table 1). Eight patients had deconditioning, the most common etiology of dysphagia. The FOIS IQR ranged from 3 to 6 (Median: 5) and the MSS from 0 to 1 (Median: 0). The IQR of PAS when swallowing Thin ranged from 1 to 7 (Median: 2), YPR-SRS (Vallecula residue) ranged from 1 to 2 (Median: 2), and YPR-SRS (Pyriform sinus residue) ranged from 2 (Median: 2) (Table 2). 

### 3.2. Overall Patient Statistical Results

PAS showed significant differences between liquids (*p* < 0.01) and was significantly lower when swallowing C-Thick than when swallowing Thin in post-hoc analysis (*p* < 0.05, Z = 2.81, r = 0.39). YPR-SRS (Vallecula residue) showed no significant differences between liquids. YPR-SRS (Pyriform sinus residue) showed significant differences between liquids (*p* < 0.01) but not in post hoc analysis. Swallowing reflex initiation showed a significant difference between liquids (*p* < 0.01) and was significantly lower in C-Thick than in Thin in post hoc analysis (*p* < 0.01) (Figure 2). The subjective difficulty of swallowing differed significantly between liquids (*p* < 0.01, Z = 0.327, r = 0.45) and was significantly lower with C-Thick than with Thick in the post hoc analysis (*p* < 0.05, Z = −2.55, r = –0.35) (Figure 3).

A significant positive correlation was observed between YPR-SRS (Pyriform sinus residue) and the swallowing reflex initiation in C-Thick (R^2^ = 0.471, *p* < 0.05) (Figure 4).

A significant positive correlation was observed between YPR-SRS (Pyriform sinus residue) and the swallowing reflex initiation in C-Thick (R^2^ = 0.471, *p* < 0.05).

### 3.3. Differences by Liquid for Each Patient

All PAS had a score of 1 on C-Thick. Unless the Thick score was 1, the patients improved their scores more when they swallowed C-Thick than when they swallowed Thick. Similarly, unless the score for C-Thin was 1, all respondents improved their scores for C-Thick rather than C-Thin.

The subjective difficulty of swallowing was felt to be easier with C-Thick than with Thick by 8 of the 13 patients (No. 1, 2, 3, 4, 5, 7, 11, 12, and 13). Five of the thirteen respondents (No. 5, 8, 10, 11, and 13) felt that C-Thick was easier to swallow than C-Thin. 

## 4. Discussion

In this study, we determined whether C-Thick is superior to the previously reported effect of C-Thin on improving swallowing. The study population consisted of elderly patients with dysphagia, and the effectiveness was tested on patients with different FOIS distribution and etiology of dysphagia. Due to the small sample size, the study focused not only on statistical results but also on individual numerical changes. As a result, changes in swallowing status were observed in all patients when swallowing C-Thick except those who did not show pharyngeal residue or laryngeal penetration when swallowing Thin.

Laryngeal penetration and/or aspiration were evaluated by PAS. The results showed that even in patients with laryngeal penetration with Thick and C-Thin, all patients no longer had laryngeal penetration with C-Thin. Regarding the effect of C-Thick on improving swallowing, we first considered the effect to be due to liquid thickening. The results of a previous systematic review recommended that Thin not be used routinely in adults with oropharyngeal dysphagia (OD) and that viscous fluids are more beneficial in reducing the risk of laryngeal penetration and/or aspiration in patients with OD [12]. However, the European Society for Swallowing Disorders reported that increased bolus viscosity increases safety during swallowing but increases the risk of post-swallowing airway invasion due to increased oral and pharyngeal residues [18]. Thus, the increase in viscosity of the fluid depends on the characteristics and severity of the individual patient with OD. In the subjects of the present study, laryngeal penetration and/or aspiration improved more with Thick than with Thin except for those who did not already have laryngeal penetration with Thin and one patient. This suggests that liquid thickening was effective. Furthermore, laryngeal penetration was no longer observed with C-Thick in the three patients who had laryngeal penetration with Thick. These results suggest that carbonation may improve swallowing.

The swallowing reflex initiation was significantly better with C-Thick than with Thin, but there was no significant difference between Thick and C-Thick. Two of the three patients who had laryngeal penetration with Thick showed no change in the site of triggering of the swallowing reflex with Thick and C-Thick. Nevertheless, since carbonated water has been reported to increase swallowing muscle activity [3] and swallowing pressure [3], it may have improved laryngeal penetration even if the site of swallowing reflex initiation was not changed.

The relationship between subjective difficulty of swallowing and penetration and/or aspiration should also be noted. All patients found it easier to swallow with C-Thick than Thin. Previous studies have reported lower palatability and lower patient preference for Thick [12,19]. The previous study also reported that sweetened carbonated beverages were subjectively easier to swallow [10]. Furthermore, Miyaoka et al. [20] studied the difficulty of swallowing samples of umami, salty, bitter, sour, and sweet tastes and reported that sweet tastes tended to be easier to swallow. Subjective swallowing difficulty is considered to be not only the ability to swallow without penetration but also a complex sensation that includes pleasant stimuli such as sweetness and carbonation to the oral cavity and pharynx. Some patients found Thick easier to swallow than Thin although there was no significant difference, suggesting that C-Thick was easier to swallow due to the addition of carbonation and sweetness. Unlike Thin, however, the bubbles in the liquid are much slower to rupture in viscous liquids and collapse inwardly [21]. Carbonated water stimulates nociceptors in the oral mucosa when carbon dioxide gas (CO_2_) dissolved in the liquid reacts with carbonic anhydrase IV (CA-IV) in salivary enzymes to produce carbonic acid (H_2_CO_3_) [7]. However, the slower the foaming of CO_2_, the less it reacts with CA-IV and the less it stimulates the oral mucosa. Therefore, the carbonic acid stimulation felt in the oral cavity and pharynx was weaker than that of C-Thin.

Nevertheless, C-Thick is superior in terms of ease of swallowing and improvement of penetration and/or aspiration, as it combines prevention of early invasion into the pharynx by Thick with carbonic acid stimulation. Patients with dysphagia who routinely take Thick have low palatability, resulting in low fluid intake and a greater risk of dehydration [22,23]. Therefore, C-Thick, with its low subjective swallowing difficulty and a compound sensation that includes palatability, can be proposed as a safe method of fluid intake for older patients with dysphagia who have complex diseases, potentially reducing the risk of dehydration.

Deconditioning was the most common etiology of dysphagia in the present patients. Causes of dysphagia due to deconditioning have been described, including decreased muscle mass due to immobilization, sarcopenia, polypharmacy, and disuse atrophy of the pharyngeal muscles [24]. The central sensory pathway, an important component of the swallowing motion that is modulated by sensory stimulation, has been reported to be well-preserved in patients with deconditioning [22]. Although some patients had cerebral infarction and Parkinson’s disease, we believe that the central sensory pathways were relatively preserved because the degree of physical disability was mild. It has also been reported that laryngeal sensation, which may be impaired in neurological disease, does not play a significant role in the carbonic acid response [24]. In subjects with neurological disease in this study, although there were residuals of the pharyngeal or swallowing reflex that was poorly elicited when the subject swallowed Thin or Thick, the central sensory pathways that transmit carbonic acid stimulation were preserved; this may be the reason the swallowing reflex improved when the subject swallowed C-Thick. No significant differences were found overall for pharyngeal residues, and the characteristics of the changes due to thickening and carbonation were not evident. Regarding the relationship with other evaluations, a correlation was found between pyriform sinus residue and swallowing reflex initiation in C-Thick. Ko et al. [25] reported that among patients with dysphagia, those with aspiration pneumonia had poor PAS, delayed swallowing reflex, and increased vallecula and pyriform sinus residue. Delayed swallowing reflex increases the risk of residual bolus in the vallecula or pyriform sinus. As some patients had improved pyriform sinus residue and swallowing reflex initiation with C-Thick compared to other liquids, the relationship between swallowing reflex and pyriform sinus residue may be more precise. The number of subjects in this study was small, and this relationship may differ in a more extensive study, including subjects with different characteristics. However, since the subjects in this study were older patients with dysphagia with complex diseases and the correlation coefficients were moderate, the results may be similar in other older patients with complex diseases.

In our study, the etiology of dysphagia varied. However, older patients often have multiple diseases, and their swallowing function declines. None of the subjects had acute neurological disease. Although the number of subjects was small, the effect sizes were moderate for the items that differed significantly in the statistical analysis. We believe that the results of this study can be applied to provide a safe method of fluid intake for many older patients with dysphagia with complex diseases. The present study could not determine the direct effect of C-Thick on pharyngeal residue. Considering that many of the patients with dysphagia in this study were deconditioned, the changes in swallowing pressure, swallowing reflex, and pharyngeal residue with C-thick should be measured in more detail in the future. In addition, a detailed study of the swallowing dynamics, safety, and efficacy of C-Thick in each phase of swallowing using the videofluoroscopic swallowing study is needed in the future.

## 5. Conclusions

C-Thick may be easier to swallow than Thick and may improve penetration and/or aspiration in older patients with dysphagia with complex diseases. Future studies are needed to investigate the swallowing behavior of the oral and pharyngeal phases in patients with dysphagia due to deconditioning in more detail.

## Figures and Tables

**Figure 1 healthcare-10-01769-f001:**
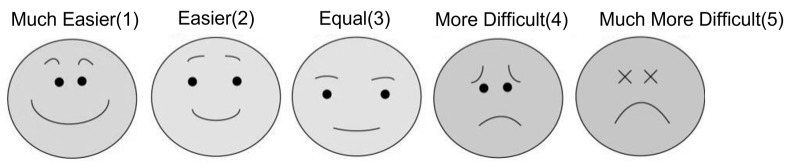
Face scale of the subjective difficulty of swallowing compared to thin liquid.

**Figure 2 healthcare-10-01769-f002:**
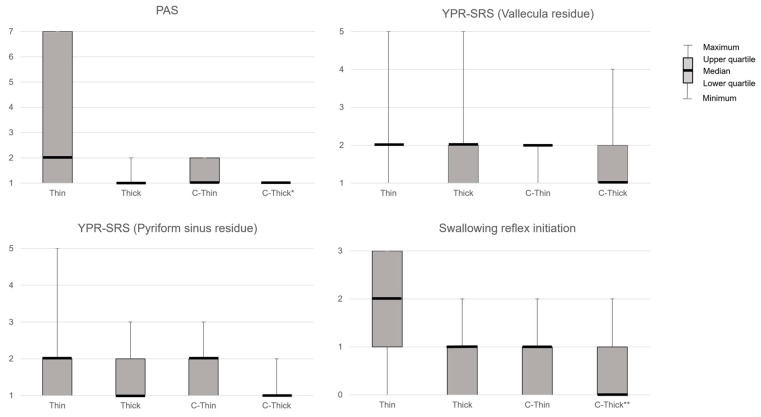
FEES results for each liquid. PAS, Penetration–Aspiration Scale; YPR-SRS, Yale Pharyngeal Residue Severity Rating Scale; Thin, thin liquid; Thick, thickened liquid; C-Thin, carbonated thin drink; C-thick, carbonated thickened drink. *, *p* < 0.05 (vs. Thin); **, *p* < 0.01 (vs. Thin).

**Figure 3 healthcare-10-01769-f003:**
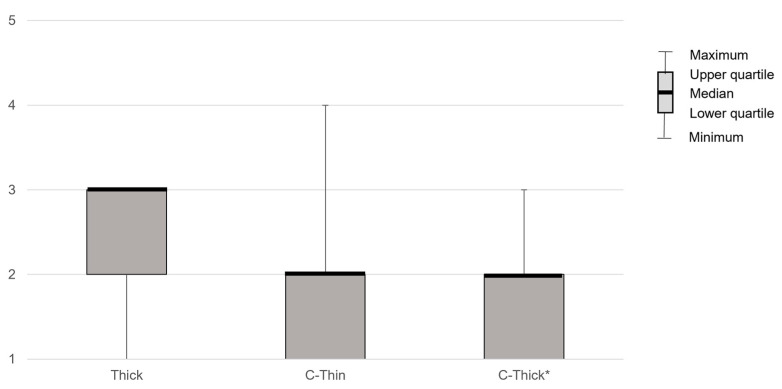
The patient’s subjective difficulty of swallowing compared to thin liquid. Thin, thin liquid; Thick, thickened liquid; C-Thin, carbonated thin drink; C-thick, carbonated thickened drink. *, *p* < 0.05 (vs. Thick).

**Figure 4 healthcare-10-01769-f004:**
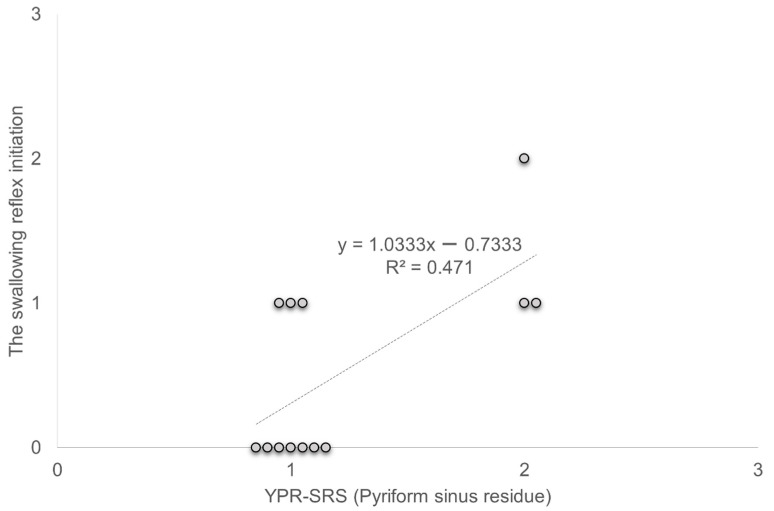
Correlation between YPR-SRS (Pyriform sinus residue) and swallowing reflex initiation. YPR-SRS Yale Pharyngeal Residue Severity Rating Scale.

**Table 1 healthcare-10-01769-t001:** Patient characteristics.

	Sex	Age	Underlying Disease	Etiology of Dysphagia	FOIS	MSS
No. 1	Male	73	Cerebral infarction Congested heart failure Dementia	Cerebral infarction	1	1
No. 2	Male	87	Gastric ulcer Chronic obstructive pulmonary disease	Deconditioning	5	0
No. 3	Male	77	Vascular dementia Diabetes mellitus	Deconditioning	4	1
No. 4	Male	89	Atrial fibrillation Dementia	Deconditioning	7	0
No. 5	Male	73	Parkinson’s disease	Parkinson’s disease	5	2
No. 6	Male	73	Cerebral infarction Dementia	Cerebral infarction	1	0
No. 7	Female	68	Parkinson’s disease spondylolisthesis	Parkinson’s disease	7	0
No. 8	Male	82	Chronic kidney failure Peripheral artery disease Hypertension	Deconditioning	2	0
No. 9	Female	64	Depression	Deconditioning	7	0
No. 10	Female	86	Chronic obstructive pulmonary disease Chronic heart failure Liver function failure Hypertension	Deconditioning	4	1
No. 11	Female	99	Chronic subdural hematoma Diabetes mellitus Aortic stenosis Hypertension	Deconditioning	5	1
No. 12	Female	82	Cerebral infarction Hypertension	Cerebral infarction	6	0
No. 13	Male	82	Chronic kidney failure Hypertension Dementia	Deconditioning	3	0

FOIS, Functional Oral Intake Score; MSS, Murray secretion scale.

**Table 2 healthcare-10-01769-t002:** Differences by liquid for each patient.

PAS					YPR-SRS (Vallecula Residue)				
Number	Thin	Thick	C-Thin	C-Thick	Number	Thin	Thick	C-Thin	C-Thick
1	2	1	1	1	1	2	2	2	1
2	2	2	1	1	2	2	3	2	2
3	1	1	1	1	3	2	2	2	2
4	2	1	2	1	4	2	2	2	1
5	7	1	2	1	5	5	5	2	3
6	7	2	1	1	6	1	1	1	1
7	1	1	1	1	7	2	2	2	2
8	1	1	1	1	8	2	1	2	1
9	1	1	1	1	9	2	1	2	1
10	7	1	2	1	10	2	1	2	1
11	7	2	2	1	11	4	2	2	2
12	1	1	1	1	12	1	2	1	1
13	2	1	2	1	13	2	3	2	4
Median	2	1	1	1	Median	2	2	2	1
IQR	1–7	1	1–2	1	IQR	2	1–2	2	1–2
**YPR-SRS (Pyriform sinus residue)**					**Swallowing reflex initiation**				
1	2	2	2	1	1	2	1	2	0
2	2	2	2	2	2	1	1	1	1
3	1	1	1	1	3	2	2	1	1
4	2	1	2	1	4	0	0	0	0
5	5	3	3	2	5	3	1	0	1
6	1	1	2	1	6	3	0	1	0
7	2	1	1	1	7	1	0	0	0
8	1	1	1	1	8	1	0	1	0
9	2	1	1	1	9	1	0	0	0
10	3	1	2	2	10	3	2	2	2
11	5	3	2	1	11	3	1	1	0
12	1	1	1	1	12	2	1	2	1
13	1	1	1	1	13	1	1	0	1
Median	2	1	2	1	Median	2	1	1	0
IQR	1–2	1–2	1–2	1	IQR	1–3	0–1	0–1	0–1

PAS, Penetration–Aspiration Scale; YPR-SRS, Yale Pharyngeal Residue Severity Rating Scale; Thin, thin liquid; Thick, thickened liquid; C-Thin, carbonated thin drink; C-thick, carbonated thickened drink; IQR, interquartile range.

## Data Availability

All data presented in this study are available in text.

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
