# Peer review of "Effects of Carbonated Thickened Drinks on Pharyngeal Swallowing with a Flexible Endoscopic Evaluation of Swallowing in Older Patients with Oropharyngeal Dysphagia"

_healthcare, 2022, doi:10.3390/healthcare10091769_

Round 1

Reviewer 1 Report

This manuscript is mainly focused on determination the efficacy of carbonated and sweetened drinks routinely used for patients with dysphagia.

There are some remarks and suggestions as follows:

1.      The title of the manuscript may be reconsidered in order to emphasize the current population studied. It could be “ Effects of carbonated thickened drinks on pharyngeal swallowing with flexible endoscopic evaluation of swallowing in elderly patients of oropharyngeal dysphagia

2.      The novelty and  significance  of the proposed manuscript  could be better highlighted

3.      The length and contents of the introduction section  could be reduced in order to be avoided the repetition of already established facts and dependencies. 

4.      It could be useful in section „Materials and methods“ to be described in more details the composition and preparation of (Thick) and (C-Thick)

5.       The small sample size and the lack of uniformity in  the etiology of dysphagia, as well as the lack of calculations of statistical effect sizes and confidential intervals could raise doubts about the significance of the study conducted

Author Response

We thank Reviewer 1 for carefully reading our manuscript and for helpful comments. Our response to Reviewer 1's comments is attached.

Reviewer 2 Report

In this manuscript (healthcare-1874093), the authors investigated the effects of carbonated thickened drinks on patients with dysphagia. The results that the carbonated thickened drink was the easiest to swallow for them are reasonable and meaningful for many caregivers of patients with dysphagia, but I have some concerns about this MS:

As the authors described in the limitations, the weak points of the study are the small sample size and the lack of uniformity in the etiology of dysphagia.

The authors described that the central sensory pathways were relatively preserved in patients with cerebral infarction and Parkinson’s disease at line 251 on page 7. But they did not show each Hyodo score of the participants in Table 1. Readers cannot judge the effects of carbonated thickened drinks on such patients because it is not clear whether they maintained swallowing reflexes or not. Table 1 should include the Hyodo score, PAS, and YPR-SRS due to the small sample size and the variety of the etiology of dysphagia.

The FOIS scores in Table 1 display considerable variation from 1 to 7. It seems to be difficult to make the patients with FOIS score 1-2 swallow liquid of 5-15cc. The authors should provide a supplementary explanation of how to maintain their safety in FEES.

Figure 4 seems to involve a high degree of risk to produce an entirely different result depending on the change of sample size. The significant correlation between YPR-SPS and the swallowing reflex initiation in C-Thick is reasonable, but it may result from a chance.

Author Response

We thank Reviewer 2 for carefully reading our manuscript and for helpful comments. Our response to Reviewer 1's comments is attached.

Round 2

Reviewer 2 Report

In this manuscript (healthcare-1874093), the authors revised the first version. These revisions may be accessible to many readers. I judge that this MS generally reaches the publication standard.